# Effective Behavioural Dynamic Coupling through Echo State Networks

**Christos Melidis** [1,*] and **Davide Marocco** [2]

1   School of Computing, Electronics and Mathematics, Plymouth University, Drake Circus,
    Plymouth PL4 8AA, UK
2   Dipartimento di Studi Umanistici, University of Naples "Federico II", via Porta di Massa 1, 80138 Naples,
    Italy; davide.marocco@unina.it
*   Correspondence: christos.melidis@plymouth.ac.uk

**Abstract:** This work presents a novel approach and paradigm for the coupling of human and robot dynamics with respect to control. We present an adaptive system based on Reservoir Computing and Recurrent Neural Networks able to couple control signals and robotic behaviours. A supervised method is utilised for the training of the network together with an unsupervised method for the adaptation of the reservoir. The proposed method is tested and analysed using a public dataset, a set of dynamic gestures and a group of users under a scenario of robot navigation. First, the architecture is benchmarked and placed among the state of the art. Second, based on our dataset we provide an analysis for key properties of the architecture. We test and provide analysis on the variability of the lengths of the trained patterns, propagation of geometrical properties of the input signal, handling of transitions by the architecture and recognition of partial input signals. Based on the user testing scenarios, we test how the architecture responds to real scenarios and users. In conclusion, the synergistic approach that we follow shows a way forward towards human in-the-loop systems and the evidence provided establish its competitiveness with available methods, while the key properties analysed the merits of the approach to the commonly used ones. Finally, reflective remarks on the applicability and usage in other fields are discussed.

**Keywords:** dynamic neural networks; mobile robot navigation; gesture recognition; behaviour dynamics; real-time action recognition

---

## 1. Introduction

Coupling the dynamics of humans' movements and the dynamics of a machine in order to control and direct the machine dynamics is a complex task. Mapping signals from one to the other in a continuous manner, in such a way that human users find intuitive and capable of expressing their own wishes and intentions, implies both detection and recognition of the input signals, as well as the full exploitation of their temporal aspects. Moreover, such detection and classification of sequences should be performed on the fly, in order to make the user in full control of the machine, therefore, a computational system that performs both tasks in real-time is of crucial importance in the field of human machine interaction.

Whether the machine is a computer, a robot, or an integrated system in which both human's and machine's autonomy are involved (i.e., a self-driving car), being able to provide a direct and natural way of interaction between the human to the machines which are in control can ease the usage of such systems, and also bring them 'closer' to the operator. 'Closer' in the sense that users do not perceive the machine as an external entity, but a continuation and expansion of their own body. At the same time, the emergence of adaptive computational techniques allows for systems

that seamlessly adapt to user preferences. Indeed, being able to connect humans and machines in such a way that the machine adapts to the user willing and intentions, rather than forcing the user to learn how to use and forcefully direct a given machine, has the potential to produce an easier, more comfortable and, above all, "natural" usage of the system [1,2]. Therefore, the approach towards a system that can adapt to the users, in order to detect and classify their actions, has an unquestioned importance in the advancement of action recognition systems, and ultimately in human-machine and human-robot interaction.

Adaptation towards the user is important [3], as it allows for personalised patterns of communication between the user and the machine. Indeed, it is shown to improve the user experience, personalised controls can also enhance the usability of the system itself, making its usage easier and more intuitive [3]. At the same time, to provide a natural way of communication between the user and the machine, the system must be able to recognise a specific sequence in a timely manner from a stream of data, effectively placing the human user in the interaction loop. The latter enables coupling the user with the robot (i.e., performing regression), rather than working under the more typical paradigm of classification.

Imagine the action of driving a car: Not only a consistency and accuracy in recognising the driving commands is needed from the on-board mechanics and electronics of the car, but also the driver's ability to perform adjustments on the steering wheel—the input for the car control system—based on the car's behaviour is equally important. Likewise, when a human and a robot are coupled in their actions and behaviour, the extent at which the user performs an input command depends on how the robot implements the corresponding behaviour. Modulating the behaviour of the robot requires that the corresponding user behaviour is effectively recognised and propagated to the control system of the robot. In the particular case of continuous interaction, being able to inform the robot of the input magnitude or intensity is also fundamental. Thus, having user and robot behaviours coupled requires the following characteristics: (i) partial input observations to yield partial output results; (ii) the input signal's intensity to be propagated to the output; and (iii) smooth transitions in the recognition of different input signals.

Another important aspect of such interaction is time. Specifically, the time required for the computations of the recognition model and for handling the dynamics of the input signals. In this context, three are the main aspects that require attention: (i) the recognition model should be able to accommodate input patterns of different lengths; (ii) it should be trained and able to adapt to different users needs and preferences in a short time, so that the user does not disengage; (iii) the recognition should be implemented with a low complexity of computation [3].

Adaptive methodologies that present useful features like the ones above have only started to appear, most of them working under a classification paradigm [4,5]. In this context, the challenges presented are mainly two: (a) detecting that a sequence is actually present in the data stream received from the input and (b) correctly classifying it. Most research features these two aspects with independent mechanisms [6–10], however, having a unified mechanism for the two tasks, saves computational resources overall and, at the same time, the recognition process becomes faster.

Moreover, the task of dynamic sequence recognition becomes especially complicated when working with real and continuous streams of data and the complexity increases when the sequences have different lengths. Methods used for the classification span from distance measures (e.g., Dynamic Time Warping) [11,12] and statistical models (e.g., Hidden Markov Models) [13,14], to artificial neural architectures (e.g., Recurrent Neural Networks) [15–19] and hybrid solutions [20]. These methods vary in complexity and adaptability, with Recurrent Neural Networks being one of the most promising direction in the field [21]. Adaptation of RNNs though, is known to have high computational complexity. In addition, the training procedure is shown to have difficulties in finding good solutions, usually referred to as a gradient vanish problem [22–24].

Given the inner complexity of the recognition task itself, working in real world environments is particularly difficult and demanding for adaptive models. Performance degrades rapidly when

working directly with noisy user data taken from real input devices, making most methods not applicable in real world situations. Cleaning and pre-processing input data, as it is often required for model to work, is not a viable option when the fundamental demand is for a method that should be readily available to the user and work reliably in real-time. The task becomes even more difficult when the input is sampled in real time and is treated continuously. Not having the ability to segment the input data, i.e., not having a starting and stopping point, makes the usage of recurrent methods necessary, as they can integrate the signal continuously in time. On the other hand, training such models requires clean data to perform well, making them difficult to train with data obtained from real users. A potential solution in this case is a computational model that is able to capture the internal dynamics of a behaviour, such as an action performed by the user on a given input device, and thus provide a robust recognition [3].

A recurrent architecture that is shown to work well with noisy data under the restrictions mentioned above is the Echo State Network approach. ESNs seems to perform surprisingly well with noisy data directly taken from a user actions and can also adapt rapidly, making their usage for user oriented systems particularly appealing [19,25–29]. In the present paper, since we are interested in behaviour recognition, data comes directly from the user manipulations of an input device. Data can be noisy and the user repetition is not always perfect, resulting to training sets of data with high degrees of noise and variation between samples (e.g., gestures, behaviours). The ESN approach followed here provides a stable and robust mapping of the input commands for user behaviour recognition.

For the investigation and validation of the method, we have followed a methodology that encompasses three stages. Firstly, we establish the validity of the proposed setup and neural architecture by benchmarking and reporting its accuracy on the recognition of actions obtained from a publicly available dataset. This allows to compare the proposed neural architecture against alternative state-of-the-art methods and also against baseline methods. Secondly, we investigate the properties of the architecture on a dataset of sequences created in house with actions recorded by the experimenter with a Leap Motion device and made on purpose to better resemble the real ones that might be obtained by casual users. The intention is to get more detailed information about the property of the system on realistic sequences before exposing it to real users. Finally, we perform a user testing of the system on a small group of people, asking them to control a simulated robot. Characteristics of the neural architecture employed, as well as methodology and results of such investigations are described in the following sections.

## 2. Material and Methods

### 2.1. Echo State Network

Echo State Networks (ESN), as seen in Figure 1, provide an architecture for efficient training of Recurrent Neural Networks (RNN) in a supervised manner [30,31]. One can distinguish two main components in an ESN. Firstly, the Dynamic Reservoir (DR), a large, random, recurrent neural network with fixed weights. These weights get initialised once and are not adapted through the training procedure. The DR is activated by the input and the feedback from the output providing a non-linear response to the input signal. The neurons of the DR usually have sigmoidal activation functions, with hyperbolic tangents to be the prevailing choice. The second part of the ESN is the output, resulting from a linear combination of the reservoir's activations. Only these weights connecting the reservoir with the output are adapted through the training procedure.

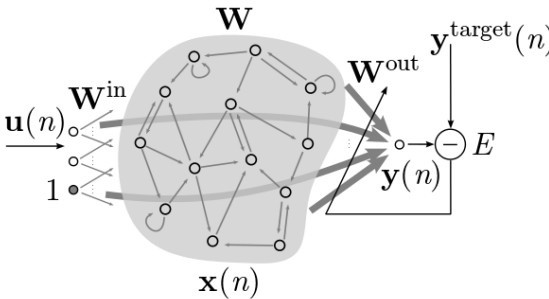

**Figure 1.** The Echo State Network architecture [32].

For an ESN to function properly, the *echo state property* (ESP) is essential. ESP states that the dynamics of the DR will asymptotically washout, from the initial conditions. It has been observed, that this can be achieved by scaling the *spectral radius* of the DR weights *W* to be less than unity [33]. That is the largest eigenvalue of the weight matrix for the DR weights should be less that unity. This condition states that the dynamics of the ESN is uniquely controlled by the input, and the effect of the initial states vanishes.

The setting of spectral radius is also associated with the *memory* of the DR [33,34]. That is the time steps it takes for the dynamics of the reservoir to washout and thus the past time steps for which information is incorporated to produce the output.

Echo State Network's Dynamics Formalisation

Assuming an ESN consisting of $N$ units in the DR, $K$ input units and $L$ output units. A matrix $W_{in}$ of size $[K \times N]$ connecting the input to the DR, a matrix $W$ of size $[N \times N]$ describing the connections amongst the DR units and a matrix $W_{out}$ of size $[N \times L]$ connecting the DR to the output, and finally a matrix $W_{outFb}$ of size $[L \times N]$ connecting the output to the DR establishing the feedback connections from the output to the DR.

Assuming time $n$, the input signal driving the reservoir is $u(n) = [u_1(n) \cdots u_K(n)]$, the state of the DR neurons is $x(n) = [x_1(n) \cdots x_N(n)]$ and the output signal is $y(n) = [y_1(n) \cdots y_L(n)]$. The state of the reservoir is updated according to

$$\mathbf{x}(n+1) = (1-\alpha)x(n)+$$
$$\alpha f(\mathbf{W}x(n) + \mathbf{W}_{in}\mathbf{u}(n+1) + \mathbf{W}_{outFb}\mathbf{y}(n)) \tag{1}$$

where $f$ is a sigmoid function usually the logistic sigmoid or the *tanh* function, in our case selected to be a hyperbolic tangent. The parameter $\alpha$ (referred to as leaking rate) regulates the percentage of the contribution of the state's previous time step to the current one. For small $\alpha$ continuous time dynamics can be approximated [35]. Setting the leaking rate to small values forces the reservoir's dynamics to a slower adaptation, in cases increasing the short term memory of the reservoir. Generally the $\alpha$ parameter can be understood as the speed of the reservoir's update dynamics discretised in time. Thus, it provides an approximation of the time interval between to consecutive -discrete- samples in the continuous -real- world. In our case the leaking rate allows for and explicit control over the memory of the reservoir, by effectively re-sampling part of the reservoir's state $x$ every time step.

The extended system state $\mathbf{z}(n+1) = [\mathbf{x}(n+1); \mathbf{u}(n+1)]$ at time $n$ is the concatenation of the reservoir and input states. The extended system state, depending on the particulars of the implementation can also include the output of the reservoir, if the output connections of the reservoir are recurrent. Here there is no recurrency in the output and thus the extended system state is as shown above.

The output signal is obtained from the network, given the extended system state by,

$$\mathbf{y}(n+1) = g(\mathbf{W}_{out}\mathbf{z}(n+1)) , \tag{2}$$

where $g$ is an output activation function typically the identity or a sigmoid, in our case the identity.

## 2.2. Training Procedure

During training the only weights adapted are the ones connecting the DR to the output, $W_{out}$. Let us assume a driving signal $u = [u(1), \ldots, u(n_{max})]$ and a desired output signal $d = [d(1), \ldots, d(n_{max})]$. The training procedure of the ESN involves two stages: (a) sampling and (b) weight computation.

### Sampling

In this stage the output is 'written' in output units, a procedure referred to as teacher forcing, and the input is provided through the input units. The network is initialised using a zero initial state $x$.

The network is driven by the input and output signals for $n$ times $n = 0, \cdots n_{max}$, at each time step having as input $u(n)$ and teacher signal $d(n-1)$, this since there exists the feedback from the output. For the first time step where $d$ does not exist, it is set to zero.

For each time step, after the washout period, the extended system states $z(n)$ and the teacher signal $d(n)$ are collected. The washout period includes those time steps just after the presentation of an input signal to the network where the systems extended states are discarded and not used in the training. This is to wait for the network to settle and the internal dynamics to stabilise and the network to settle to the input provided.

The extended states are collected in a matrix $S$ of size $[n_{max} \times (N + K)]$ and the desired outputs $d(n)$ in a matrix $D$ of size $[n_{max} \times L]$.

Now, the desired output weights $W^{out}$ can be calculated as follows. First, the correlation matrix of the extended system states is calculated, $R = S'S$. Then, the cross-correlation matrix of the extended states against the desired outputs $d$, $P = S'D$. Finally, the calculation of the output weights of the network $W_{out}$ is done by calculating the pseudoinverse of S, $S^{\dagger}$,

$$W^{out} = (S^{\dagger}D)' \tag{3}$$

## 2.3. Intrinsic Plasticity

Selecting the spectral radius of the reservoirs weight matrix is one of the most important parameters while using dynamic reservoirs. Intrinsic Plasticity (IP) provides an unsupervised method for the adaptation of the Dynamical Reservoir [36,37]. The idea is that the activation functions of the neurons are adapted to fire under a certain, usually exponential, distribution. This results to sparse activations of the reservoir neurons, with each one capturing only important features of the input signal. The IP rule is local in space and time and aims at maximizing input to output information transmission for each neuron.

In our case, where the training data is noisy, IP is show to alleviate the overall performance helping in the decorrelation of the noisy input signals in the training procedure.

Using a hyperbolic tangent as an activation function for the reservoir's neurons, the intrinsic parametrisation can be derived by adding a gain $a$ and a bias $b$, to the activation function $f'(x) = f(ax + b)$ and now working with f' as the activation function of the reservoir's neurons. Then, the online adaptation rule of IP according to [38] is derived to be,

$$\Delta b = -\zeta(-\mu\sigma^{-2} + y\sigma^{-2}(2\sigma^2 + 1 - y^2 + \mu y)) \tag{4}$$

$$\Delta a = \zeta a^{-1} + x\Delta b \tag{5}$$

where $\zeta$ is the learning rate for the IP, $\mu$ the mean of desired activation distribution and $\sigma^2$ it's variance. All signals, $x, y$ and parameters $a, b$ are of the same time step $n$.

### 2.4. Parametrisation of the System

The matrices $W_{in}$ and $W_{outFb}$ have 10% connectivity and are initialised in ranges $[-0.9, 0.9]$ and $[-10^{-4}, 10^{-4}]$ respectively. The DR matrix $W$ has a 20% connectivity and is adapted through the IP rule, needing no explicit *spectral radius* setting. The parameter $\alpha$ in state calculation is set to 0.5 for both training and usage of the network. The size of the DR was chosen to be $N = 128$.

For the IP learning rule, the learning rate is $\zeta = 0.0001$, the mean $\mu = 0.0$ and the variance $\sigma^2 = 0.8$. The gain parameter $a$ is initialised to unity, while the bias parameter $b$ to zero.

For the adaptation of the ESN the training sequence is presented to the network and the IP rule is applied according to Equations (4) and (5). Then the sequence is presented once more and the collection matrices $S$ and $D$ are created and the output weights $W_{out}$ are calculated as described above in Equation (3).

The optimal configuration was achieved by repeated experiments, although there exist methods for automatic or semi-automatic fixing of the parameters. The autonomous adaptation of the reservoir through the IP rule allows for a variability in the setting of the parameters, since the reservoir neurons are adjusted to have a maximal information transfer for the given input signal.

## 3. Experimental Setup

### 3.1. Technical Details

For the testing of the system a Leap Motion sensor was used, as seen in Figure 2. The system is initialised as described above for the input device. The ESN architecture described above was coded in Python using Theano [39]. A client-server model was implemented to provide the connection between the input device and the learning algorithm (i.e., ESN).

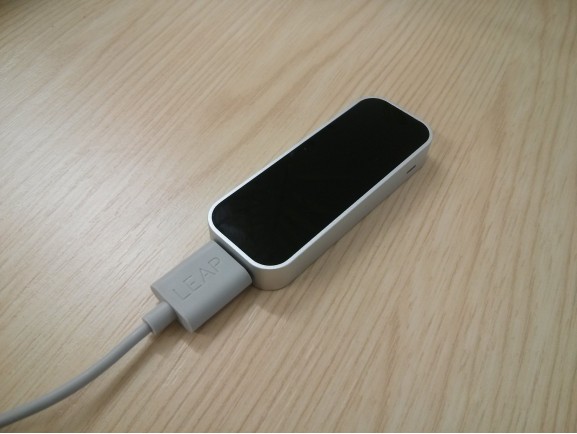

**Figure 2.** The Leap Motion input device used in the experimental setup.

### 3.2. Input Signal

The Leap Motion device is a sensory device providing tracking and skeleton data for hand and fingers positions in space. Using the *JavaScript* library provided by the manufacturer, and the client-server setup described above, we recorded six, 6, values to describe the hand position at each frame. The values recorded represent the 3 rotational and 3 translational DoF of the centre of the palm of the hand. The setup allows us to stream the input signal through the network at the sampling rate of the device i.e., >30 fps. A frame rate as high as 100 fps was able to be produced, just for testing purposes of the setup.

There is no sub-sampling performed, nor for the training set acquisition nor for the testing phase. The device is sampled at each time step, and the sample is directed to the server side where it is fed to the ESN. The ESN provides an output for each time step, recorded and used for the analysis of the performance of the system in the results section.

### 3.3. Testing Cases

For the testing of the proposed system we have worked as follows:

- We tested the validity of the proposed method on a publicly available dataset, the Cornell Activity Dataset (CAD-120) [40]. This, in order to assess the quality of the work presented and to provide evidence of the generalisation capability and flexibility of the proposed method. Although CAD is rather distant to the application field of the proposed method, it allows for the comparison of our method against a baseline, while also shows the applicability of the setup regardless the type of input signal. The results of our method are reported and compared to alternative state-of-the-art computational methods.
- In order to investigate the system in more detail, an extra set of sequences was recorded by the experimenter using the Leap Motion as input device. This has produced a new dataset on which the proposed system has been further tested, labelled in this work as (*Dataset Testing*). I this way, we are able to test the system with input sequences more applicable to our specific interest of robot control and also highlight general characteristics of the system.
- A small number of users were asked to control a simulated robot, visible on the screen of a computer, using the proposed system and the Leap Motion as input device. We refer to this test within this work as *User Testing*. In this phase of the testing, the users were asked to perform gestures using the Leap Motion device, in relation to behaviours of the simulated robot shown by the experimenter on the computer screen. It is important to mention here that the users were not instructed on the kind of gestures they should use in order to control the robot, allowing them to freely manipulate the input device at their own preferences. This resulted to different gestures being used by the users in relation to the same robot behaviour shown to them. This fact indicates the flexibility of the proposed system in personalising the control sequences and the associations between the user's gestures and the robot behaviours. Since the gestures performed by the users were different for each one of them, based on their preference, only the accuracy of the system is reported under this setup. Once the ESN was trained with the gestures performed by the users, they were asked to control the robot using their own provided gestures.

#### 3.3.1. CAD-120 Testing

The CAD-120 is a publicly available dataset with the recording of skeleton data of 10 daily activities: *making cereal, taking medicine, stacking objects, unstacking objects, microwaving food, picking objects, cleaning objects, taking food, arranging objects, having a meal*. These activities are performed by 4 different people and each is repeated 3 or 4 times. For each person and repetition, a time series of the skeleton data is used as input for the network. Although the dataset offers a confidence value for the skeletal data at any point, all have been used regardless, since ESN are known to work well with noisy data. For the reporting of the accuracy of the method on the CAD-120 dataset, a leave-one-person-out cross-validation scheme is used as found in the literature [40,41]. Given the application domain—iterative robot control—from the dataset only the skeletal data were used as input (i.e., avoiding ground truth labels and objects in the scene), including erroneous entries. The accuracy is reported as the mean of the respective accuracies for each person. Following the accuracy reporting scheme found in literature, for each activity presented to the network the readouts were averaged for the whole length of the sequence.

#### 3.3.2. Dataset Testing

In order to test the system in a setup more similar to its intended functionality the *Dataset Testing* was created in house by the experimenter. This dataset consists of 7 generic dynamic hand gestures performed using the Leap Motion device. In this case a different measure is used to calculate the accuracy, in order to highlight the mapping paradigm under which the method is used. We report the percentage of time the network output is indicating the correct input sequence presented, since

we assume no segmentation of any sort of the input sequences, neither logical (e.g., by performing a moving average of the output), nor physical (e.g., by removing the hand from the Leap Motion's recording area before and after the performance of the gesture). The Leap Motion was selected for the testing as its larger input size is more demanding for the system. Each input gesture was repeated 3 times, the system was trained using two sequences out of the three and tested on the third, unseen, one. Thus, the accuracy is reported based on a 3-fold cross validation methodology. Each gesture of the training set includes the preparation, the nucleus, and the retraction of the gesture without tagging any of those moments [42,43]. That is, each gesture includes the positioning of the hand within the device's receptive field and its removal. There was no care whether each execution of gestures was starting from the same point, nor that it had the same time span, nor that it was performed in the same manner, so to follow the exact same shape every time (e.g., performing a clockwise rotation of the same radius for the 3 times). This has been done in order to account for spatial and time variability between the input sequences.

The actions performed by the experimenter with the right hand within the range of the Leap Motion are the following:

**Push** The hand moves forward from the centre of the receptive field in the horizontal plane;
**Pull** The hand moves backwards from the centre of the receptive field in the horizontal plane;
**Swipe-right** Repeated swipe movements from the centre to the right of the receptive field in the horizontal plane;
**Swipe-left** Repeated swipe movements from the centre to the left of the receptive field in the horizontal plane;
**Clockwise Circle** the hand moves repeatedly clockwise in a circle within the receptive field in the vertical plane;
**Anti-Clockwise Circle** the hand moves repeatedly anticlockwise in a circle within the receptive field in the vertical plane;
**Up-Down** the hand moves up and down within the receptive field in the vertical plane.

It is important to note that sequences varies in length, as seen in Table 1, which presents the 7 gestures recorded for the testing together with their length. Furthermore, it is also interesting to note that for the system to be able to discriminate between sequences 4, 5, and 6 it should be able to follow their ongoing dynamics. In fact, the higher and lower hand position in gesture 6 can also be found in gesture 4 and 5, as they are part of the circle described by the hand on the vertical plane. Similarly, gestures 2 and 3 share some of the hand positions with gestures 4 and 5, since the leftmost and rightmost points also belong to the circle described by the hand on the latter gestures.

**Table 1.** The 7 gestures recorded for testing. Their description is provided in the text. The sequence length is given in frames captured by the input device.

| Description | ID | Sequence Length |
|---|---|---|
| Push | 0 | 150 |
| Pull | 1 | 195 |
| Swipe right | 2 | 144 |
| Swipe left | 3 | 129 |
| Clockwise Circle | 4 | 225 |
| Anti-Clockwise Circle | 5 | 147 |
| Up-Down | 6 | 147 |

### 3.3.3. User Testing

In this final stage of the testing, eight participants were asked to perform gestures that they would deem appropriate in order to control 4 simple robotic behaviours shown to them with a simulated mobile robot. The robot selected for this test was a simple 2 D.o.F. differential drive mobile robot. That is, 2 drive wheels are mounted on a common axis and each wheel can independently be

driven either forward or backward. A set of 4 behaviours were implemented on the robot: Forward, backward, clockwise rotation and anticlockwise rotation. The users were asked to perform their own set of input signals for these behaviours and then control the robot in a continuous fashion using their own generated signals. The system does not require that the users segment their input gestures, with transition between the input sequences being handled by the network's dynamics autonomously. The recording of the gestures was done in the same fashion as described in the previous section. The only relevant difference was that each gesture was performed only once for the training of the ESN. The participants were also asked, at the end of the testing, whether they realised any lags in the executions of the commands they sent to the robot.

In all cases where the system was used, each time point of a gesture performed is recorded, placed in a bucket and labelled with an index at the allowed frame rate of the Leap Motion. Once all gestures are performed, the network is trained, following the procedure described in Section 2.2. The machine used for the training and testing of the system, in both test cases, was a mid-range laptop with an Intel Core i5-3340M CPU @ 2.70GHz × 4 (2 cores, 4 threads), with 3.7GB of RAM and without the use of any GPU acceleration methods. The training procedure took less than a second <1 s in all cases, even for the larger testing set.

## 4. Results

Results are split in three sections for the three test cases used. First the results from the *CAD-120 Testing* are reported, followed by the *Dataset Testing* and finally the results from the *User Testing*.

### 4.1. CAD-120 Testing

In Table 2 the accuracy of the proposed method is reported. The skeletal recordings for each activity are used as input, while at the same time, all the available data regardless of their corresponding confidence value are used. The confidence value is available in the dataset and reports whether the given skeletal pose at a given frame is valid or not. For comparison the best results found in bibliography are reported for which the same input was used, that is, skeletal data without ground truth labels, or objects in the scene.

**Table 2.** Accuracy of the method on the Cornell Activity Dataset (CAD-120). * Koppula et.al reports on results with information about the objects in a scene.

| Method | Accuracy (%) |
|--------|--------------|
| [44]   | 70.2         |
| [40] * | 75.0         |
| ESN    | 73.5         |

Although not many research reports classification results excluding objects in the scene, we can observe that the method presented here is able to achieve comparable performance.

At the same time, results show how the ESN architecture is able to handle the vastly different lengths of the recorded activities in the CAD-120, which varies from 150 to 900 frames.

In addition, the unsupervised adaptation of the reservoir through the IP rule, allows for a parametrisation of the network specific to the input sequences. Indeed, we observe that, because of the IP rule, a much smaller reservoir of only 128 neurons can be used, compared to the 300 units reservoir reported in [41]. We believe this is possible thanks to the IP rule, by which the activation function for each neuron is adjusted to maximise the information transfer. Furthermore, locality in time and space makes the adaptation computationally efficient [37].

### 4.2. Dataset Testing

After running the ESN according to the training procedure described, the system was always able in every case to converge and to find the right set of output weights for the task. Once the system is

trained, sequences are then presented in random order to test the accuracy of the training. The network provides a response (output) for each time step an input is provided. Comparing the output with the gesture performed, we measured an accuracy of 87.8% for all the gestures performed. That is, 87.8% of the time steps an output was generated, it was indicating the correct gesture. It is worth to note here that this measure cannot reach 100% accuracy, since the ESN needs some time to stabilise its output for the input signal. For a more stable measure, the output of the network should have been segmented and observed only after the stabilisation. However, since we do not want to use any arbitrary set of parameters to judge the stabilisation point, we proceed with this holistic measure in the reporting of the results. Comparable performances, using less gestures, have been reported by Weber [45]. It is to be noted, however, that in Weber's work gestures have a starting and ending points, that we have not included in our work, to avoid any arbitrary interpretation of the gestures.

In Table 3 a more detailed representation of individual results obtained for each sequence are presented. During testing, each gesture is recognised during the exhibition. What we present in table is the average of correct recognitions for all time steps each pattern is presented to the network. Since the system is meant to provide a continuous output for every point of the sequence provided in input, we measure the percentage of correct recognition in time, as the input sequence is presented to the ESN. It can be seen as a measure of the correct mapping between input and output in time.

**Table 3.** Training and testing accuracy scores for the 7 gestures. Score is measured as the mean of recognised time steps for each gesture.

| Description | Training | Testing |
|---|---|---|
| Push | 0.99 | 0.99 |
| Pull | 0.99 | 0.99 |
| Swipe right | 0.99 | 0.98 |
| Swipe left | 0.99 | 0.72 |
| Clockwise Circle | 0.99 | 0.89 |
| Anti-Clockwise Circle | 0.99 | 0.70 |
| Up-Down | 0.96 | 0.88 |
| Mean | 0.98 | 0.87 |

*4.3. User Testing*

As a final step of the testing phase, the system was finally exposed to users. That is, eight people were asked to use the system and control the simulated wheeled robot without a specific task. Their only goal was to control the robot in the way they wanted.

Results in this case are very similar to the ones observed with the *Dataset Testing* condition. The ESN was able to find the right set of output weights for all sequences provided by all height users every time. Although the input sequences recorded by the users where completely arbitrary and very different in terms of overall length and gesture patterns, the proposed architecture was able to cope with the incoming signal and mapping it to the output. Notably, the overall training performance was significantly increased with respect to the previous testing, since the network had only to distinguish between four input patterns, i.e., the four gestures associated to the four pre-coded movement of the robot.

Table 4 shows the lengths of the 4 recorded input patterns (gestures) from the participants. From the table we can observe the high variability of the gestures in term of length of the input and to also, therefore, highlight the capability of the setup to deal with different lengths. The input device was sampled at the maximum allowed frame rate (i.e., 100 fps) with the length of each sequence being the number of frames recorded from the user.

**Table 4.** The length of each of the 4 gestures used by each participant are presented. The table displays the variability of the length of the gestures. Fluctuations of the gesture lengths are observed between participants and between gestures. We can clearly observe that there is no particular tendency in terms of the lengths of the selected input sequences. The length is measured by the frames recorded for each sequence, with the average frame rate of the device being 100 frames per second.

| Participant | G1 | G2 | G3 | G4 |
|---|---|---|---|---|
| P1 | 2613 | 841 | 975 | 1142 |
| P2 | 210 | 180 | 192 | 121 |
| P3 | 721 | 619 | 360 | 701 |
| P4 | 205 | 409 | 384 | 602 |
| P5 | 187 | 155 | 68 | 101 |
| P6 | 207 | 128 | 203 | 266 |
| P7 | 604 | 614 | 436 | 596 |
| P8 | 241 | 521 | 522 | 492 |

Furthermore, in order to provide a qualitative appreciation of the variability observed between user gestures, Figure 3 shows the training set (i.e., the recorded sequences made by the Leap Motion device) of 3 users depicted within the 3 respective graphs. Each figure shows 6 lines representing the 6 D.o.F of the input device during the recording of the user. Those recorded values are then fed to the ESN as input. In each figure, the separation of the four sequences, representing the four gestures made by the user, is indicated above the graphs with the label G1, G2, G3, and G4 respectively, referring to the forward, backward, clockwise, and anticlockwise movements of the robot. By visually comparing the patterns for the three users it is possible to appreciate their differences, both *within* the same user and *between* users. As already seen in Table 4, the differences in length of the sequences are noticeable. Moreover, it is also possible to appreciate the difference in which users have decided to associate their gestures to the four robot behaviours. Some users preferred periodic movements for all their input sequences (e.g., *P6*), while others chose more stable and non-periodic movements (e.g., *P8*). At the same time, as shown in *P7*, the system was also able to handle cases were periodic and non-periodic input behaviours were mixed by the user.

After the test, each user was asked to respond to a questionnaire, in order to investigate the quality of the interaction and the feasibility of the methodology proposed. It is to be noted, in fact, that given the characteristic of the task presented the subjects, it is not possible to disentangle the input sequences performed by the users at run-time with the corresponding output and isolate the single gestures recorded during the training phase by the users itself, in order to make a comparison. This makes impossible to assess the accuracy of the network in the same way as it was done for the *Dataset Testing* condition. This is also the reason behind the creation of the *Dataset Test*, i.e., to have tangible and quantitative proof of the actual works of the ESN.

Besides the specific analysis of the responses, which is not central for this work, all of the users did not report any delay in the systems response. Seven subjects out of the eight reported that they felt in control of the robot by using the Leap Motion device. This indicates that the ESN was able to map their input signals in the corresponding robot behaviours, as they were expecting. Also, all users reported that the network's training time was short, most of them having not noticed it, and the training procedure short enough, having to perform only one repetition for each control signal.

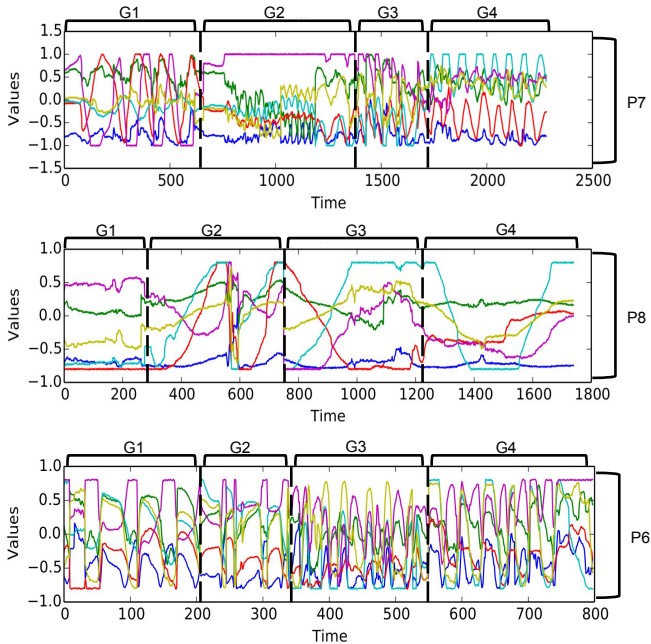

**Figure 3.** Three examples of four gestures, input behaviours, from participants P6, P7, and P8 used for the training of the Echo State Networks (ESN). The four different gestures are labelled with G1, G2, G3 and G4 at the top of each figure. Lines represent the 6 dimensions of the input signal of the Leap Motion device, plotted against time. It is possible to appreciate: (a) the visible differences in the quality of the input sequences and (b) the different lengths in time. That is, the different span along the x-axis.

## 5. Properties of the Echo-State Network and Human-Machine Interface

During the running of the ESN and the tests that have been performed, a number of observations have led to a more detailed investigation of some aspects that represents particularly interesting features for the field of human-machine and human-robot interaction. Such interesting features that the proposed ESN shows regards the way in which it solves problems concerning the variability of the length of the patterns to be classified, the complexity posed by the real-time processing of the input streams and the huge amount of noise, which is typical of the raw data that we use as input for the system.

From the same perspective, the next sections present some of the properties that we have discovered by analysing the trained ESN. Those properties, together with the above features, we believe can have an interesting impact in the way in which a system like the one presented here can shed new lights on the construction of flexible interfaces between human and machines.

### 5.1. Variability in Pattern Length

As mentioned in Section 4 the training patterns varied in length. This is a characteristic of all actions and behaviours performed by humans in real life and is also evident both in the CAD-120 dataset and in the dataset created by ourselves. Therefore, this is a fundamental problem that a human-machine interface has to face. Indeed, not confining the patterns to be of equal time scales, which will be artificial, allows for greater freedom for the user and enhances the robustness of a system trained on raw user data. The immediate benefit is that there is no need for explaining to the user the way in which the interface works. At the same time, the user will be free to behave in natural and intuitive way.

The high degree of recurrency within the DR allows for temporal dynamics of different time scales to be recorded and retained, without any explicit specification on the duration of the patterns. Both the number of neurons and the spectral radius of their connecting weights accounts for the memory size of the DR. In this work we have shown that it is possible to adapt those parameters in an unsupervised

manner using the IP rule, allowing for the DR to adapt to the input behaviours. In this way we can obtain at the same time a general architecture capable of recognising sequences of different lengths and a specific adaptation towards a specific training set provided by the user.

### 5.2. Continuous Mapping from the Raw Data Input

Dynamic actions, such as hand gestures, generally contain three phases that overlaps in times: Preparation, nucleus, and retraction [42,43], of which the nucleus is the most discriminative. The setup proposed is able to capture the discriminative part of a gesture without any explicit instructions about its location within the overall gesture performance. Thanks to this feature, transitions between gestures can be handled autonomously by the ESN. In turn, user input sequences do not need to be artificially and purposefully segmented, allowing for the continuity and natural flowing of the input to be preserved in the ESN output. It is this property of the setup that allows for the user to be placed in the loop of the controlled machine, or robot in the case of this work.

### 5.3. Geometrical Properties of the Input

Figure 4 shows a detail of three patterns from the condition *Dataset Testing* correctly recognised by the ESN:*CW*, which stands for a clockwise circle pattern performed by the user, *ACW* an anticlockwise circle pattern, and $Up - Down$, an up and down movement of the hand of the user, as recorded by the Leap Motion device. By observing the first segment of the graph and delimited by the first vertical line in the figure around *Time*100, we can see that the network correctly recognises a *CW* gesture in input (the *Value* of *CW* in the figure reaches 1.0). Interestingly, the *ACW* gesture at the same time shows a negative value. This observation can be explained by the fact that the *ACW* pattern is 'opposite' to the *CW* patter. Therefore, it suggests that geometrical properties of the input are retained and the spatial relationship between the two signals is captured and embedded from the network in its output signals.

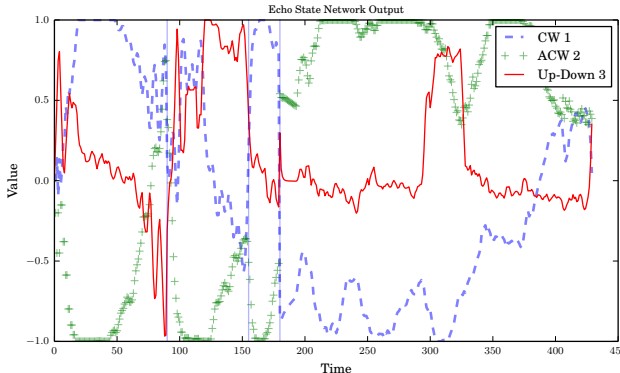

**Figure 4.** Usage of a trained ESN. The plot highlights how the geometrical properties of the input sequences are retained on the output of the network. The antagonistic behaviour between *Clockwise* (CW) and *Anticlockwise* (ACW) behaviours is shown, while the *Up-Down* motion recognition remains unaffected. In the graph the output of the network is depicted, with each colour and line style representing a pattern recognised in the input. All values are plotted against time.

Similarly, in the next section the input behaviour changes from *CW* to a mixture of both *CW* and $Up - Down$, and ultimately to just $Up - Down$ around *Time* 130. This transition is also reflected to the output of the network, but, besides the two behaviours being mixed, *ACW* remains always negative and opposing the values of *CW*. This observation indicates that, although it is possible to mix behaviours, it remains impossible to do so with geometrically opposite ones. This is an interesting feature that, to our knowledge, cannot be found in other models. Similar dynamics can also be observed in the following section of the pattern, where the $Up - Down$ recognition settles around 0, the user is performing a *ACW* gesture and, as expected, the opposite *CW* pattern is negative.

Such observation indicates that geometrical properties of the input are propagated to the output. In our example, indeed, the clockwise and an anti-clockwise motion inhibit each other. By assuming that the network has been trained to control a moving robots, it is possible to grasp the importance of this feature. For example, lets assume the *CW* motion is mapped to the robot moving forward and the *ACW* backwards. Having opposite behaviours being interpreted as 'opposite' by the system, it provides the network with an 'insight': The user cannot perform two opposite behaviours at the same time, but it can perform the $Up - Down$ gesture in combination with any of the above. At the same time, the fact that the two behaviours are opposite is also maintained in the output. Assuming that the robot behaviours are combined in a linear fashion based on the network outputs, the recognition of 'move forwards' implicitly means for the system that 'move backwards' will hold opposite values (and negative in the specific implementation presented here).

### 5.4. Recognition Before the End of the Sequence

An important feature of the system presented in this work is that it provides the correct classification before the input sequence is completed. Given the feedback from the output is fed to the reservoir, the network is able to stabilise its dynamics and recognise a given pattern at an early stage of its presentation. This feature allows the system to have a fast response to the sequence in input, making it appealing for real time control cases.

Figure 5 shows an example of the recognition of the *pull* (ID 1) sequence of the dataset. Similar behaviour is also shown by the network for the other sequences as well. That is, the network is able to classify the sequences before their completion. The time steps required for the network to settle to a sequence can vary. This is expected as the sequences do not share the same length.

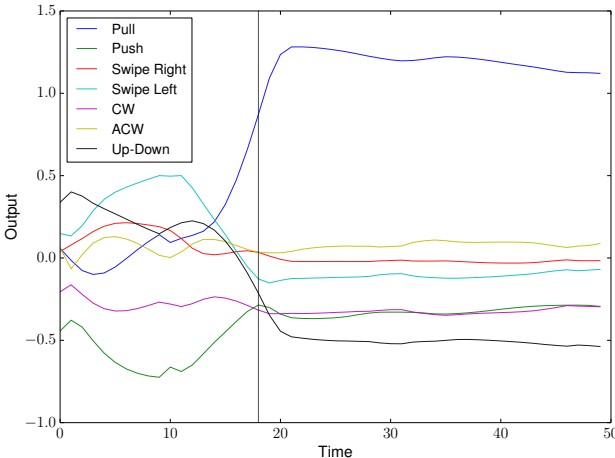

**Figure 5.** The figure shows the recognition of the *pull* (ID 1) input sequence from the initial presentation until the stabilisation of the network's output. In the graph the output of the network is depicted, with each colour and line style representing a pattern recognised in the input. All values are plotted against time.

For a more comprehensive way of how the proposed architecture captures the dynamics of the input sequence, we tested the recognition with partial input sequences. Each input sequence was used to artificially create four new sequences, each one having 25%, 50%, 75% and 100% of the original sequence. Each sequence resulted from the initial one having the same starting point but a shorter time span, by omitting the remaining elements of the sequence. In this way, a set of 28 sequences were used for testing. The accuracy of the network is measured in the same fashion as before, reporting the percentage of time the network output indicates the correct input. Each sequence was presented to the network independently, resetting the network in between the sequences presented. At the same time, sequences were shuffled so as to eliminate any of their dynamics to be retained in the ESN's reservoir.

Figure 6 shows the results obtained by the test. From the bar chart it is possible to observe that the network produces the correct answer even from the initial 25% of some sequences (i.e., *IDs*1 and 2). At the same time, for most sequences it reaches a good performance with only half (50%) of the sequence being presented. When the 75% of the input sequence is presented the network is able to recognise all input patterns with a high level of accuracy, with the exception of pattern 3, which is the only one that reaches its maximum recognition rate only when the entire 100% of the pattern is presented.

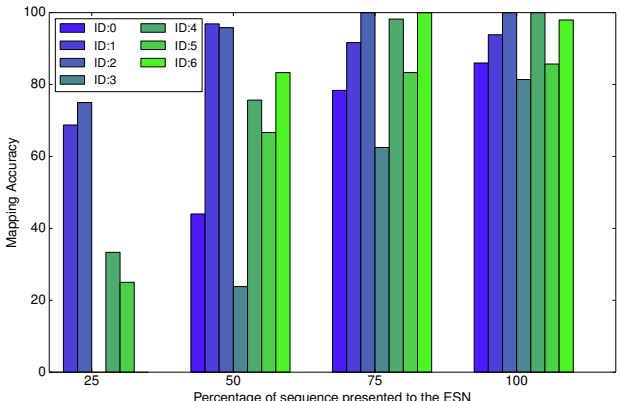

**Figure 6.** Accuracy of the ESN for partially observed inputs. Each colour represents an input sequence. The bars are grouped in four categories, each one representing the percentage of the signal presented to the network.

Given the unified structure for the detection and recognition provided by the ESN, input patterns are detected before their completion. This allows for fast responses from the system, a feature necessary for real time control. It is shown that humans are very sensitive to the response time of user interfaces, with lags greater than 100ms perceived as annoying [46,47]. Being able to provide feedback within the time span of a given input sequence, that is, during the execution of a gesture, is a challenge that ESN can achieve given the simplicity of the computations performed, which allows for very fast computation in comparison with other methods. Indeed, more complex classification systems perform even more costly computations with similar performances [15].

## 6. Conclusions

In this paper an echo-state neural architecture for the recognition of continuous time signals is presented, together with a methodology for fast and efficient training. The proposed system is tested under two different paradigms. One to analyse its properties and one to test its real world applications. Through the testing useful properties are highlighted, analysed and their potentials are discussed. Under the scope of human-robot and human-machine interaction the system's applicability is discussed. At the same time the properties of the architecture are discussed independently, in order to allow and encourage usage of the method in other fields.

The findings of this paper show that pattern recognition in continuous time signals is possible without the computational or algorithmic complexity of methods used so far in the field. The particular time signals considered here are coming from the manipulation of input devices within a human-machine interaction framework. The mapping that the proposed architecture provides was tested under a robot navigation task. In the field of robotics such an adaptive mechanism is shown to provide a just-in-time solution for a user centric system, capable of coupling the user's and robot dynamics in real time.

In the field of assistive robotics, ESN can provide a fast and reliable way of adapting the system to the users preferences. This may accommodate cases of increased of decreased mobility and the

usage of unorthodox input devices. Being able to capture, train and recognise user behaviours from their preferred input method can be alleviating for use cases that cannot be taken into account in the design procedure.

**Author Contributions:** C.M.: Conceptualization, data curation, software and writing original draft. D.M.: Conceptualization, writing, review and editing the manuscript.

**Conflicts of Interest:** The authors declare no conflict of interest.

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
