# Peer review of "Effective Behavioural Dynamic Coupling through Echo State Networks"

_applsci, doi:10.3390/app9071300_

Round 1

Reviewer 1 Report

This manuscript presents an Echo State Network to connect user inputs from a Leap Motion device to a simulated robot.  The idea seems promising towards reducing the gap in human-to-machine interfaces.  However, this reviewer has several concerns:
- Although the work is well motivated in the introduction, the "Material and methods" requires a bit more detail in the explanations of the Echo State Network.  Maybe some graph/figure could help readers to understand the concepts better.
- It is not clear how the results from the CAD-120 dataset can be extrapolated to the use case presented by the authors.  
- The acquired dataset is constrained to a reduced set of movements.  Does it affect the quality/accuracy of the classification?  Maybe authors could increase the dataset in order to provide a more general view.
- When analyzing the results obtained with the CAD-120 dataset (section 4.1), the authors use references that are a bit outdated.  Is there any other reference that could add value to the comparison with state-of-the-art alternatives?
- In line with the previous comment, have the authors considered using other approaches (see [1], although the reference is also a bit outdated) for a wider and maybe better comparison?  
- Keeping unexperienced readers in mind, it could be useful to explain how results are presented in certain figures (e.g., Figures 3 and 4).

[1] M. V. Liarokapis, P. K. Artemiadis, P. T. Katsiaris, K. J. Kyriakopoulos and E. S. Manolakos, "Learning human reach-to-grasp strategies: Towards EMG-based control of robotic arm-hand systems," 2012 IEEE International Conference on Robotics and Automation, Saint Paul, MN, 2012, pp. 2287-2292.

Author Response

We would like the thank the reviewer for their comments. Below we provide a point-by-point response to the reviewers comments and suggestions.

- Although the work is well motivated in the introduction, the "Material and methods" requires a bit more detail in the explanations of the Echo State Network.  Maybe some graph/figure could help readers to understand the concepts better.

- A figure has been added in the Materials and Methods section in line with the reviewer’s comment.

- It is not clear how the results from the CAD-120 dataset can be extrapolated to the use case presented by the authors.  

- The CAD-120 is used to establish the accuracy of the recognition capabilities of the model and as such it is used as a benchmark method.

This is described in section 3.3, “We tested the validity of the proposed method on a publicly available dataset, the Cornell Activity Dataset (CAD-120). This, in order to assess the quality of the work presented and to provide evidence of the generalisation capability and flexibility of the proposed method. Although CAD is rather distant to the application field of the proposed method, it allows for the comparison of our method against a baseline, while also shows the applicability of the setup regardless the type of input signal. The results of our method are reported and compared to alternative state-of-the-art computational methods.”

- The acquired dataset is constrained to a reduced set of movements.  Does it affect the quality/accuracy of the classification?  Maybe authors could increase the dataset in order to provide a more general view.

- The set of movements is reduced based on the user. The user decides the number and type of movements. It is also tied to the number of movements of the robot. This is the main reason for proving a testing of our method with the CAD-120 dataset, to demonstrate the recognition capabilities of the network with a publicly available dataset.

- When analyzing the results obtained with the CAD-120 dataset (section 4.1), the authors use references that are a bit outdated.  Is there any other reference that could add value to the comparison with state-of-the-art alternatives?

- Unfortunately there have not been any more recent reports on the CAD-120 than the ones provided.

- In line with the previous comment, have the authors considered using other approaches (see [1], although the reference is also a bit outdated) for a wider and maybe better comparison?  

- To support the applicability and better comparison of the architecture with existing systems, we benchmark the system with CAD-120. Being a public dataset there is a plethora of methods applied to the dataset, as it can be seen here http://pr.cs.cornell.edu/humanactivities/results.php. From these methods we report the results of relevant implementations only i.e. the ones with the same constraints.

- Keeping unexperienced readers in mind, it could be useful to explain how results are presented in certain figures (e.g., Figures 3 and 4).

- The captions of the figures have been re-examined and some changes have been made to accommodate for unexperienced readers.

Reviewer 2 Report

The paper proposes an adaptive system based on Reservoir Computing and Recurrent Neural Networks able to couple control signals and robotic behaviors.

References are rather old, mainly from, 2007 to 2013. From 2015 to 2018 (last four years) there are only 4 journal references out of a total of 39 references. Authors are encouraged to find more current works (of high-quality journals if possible) as it seems that their approach is not compared against current works.

Regarding Cyber-Physical systems, the term is included in the title, as an important aspect considered in the paper. However, the term is never introduced in the manuscript text. Authors should base their approach in CPS or either remove this term from the title, as it is confusing.

The paper proposes an adaptive architecture in the title. However, I cannot see in which part of the text is the architecture proposal. What is your concept of architecture? Doesn’t it involve any architecture diagram (functional architecture)?

The experimental setup and result evaluation are correct and well-performed.

A similarity check by using the Turnitin tool results in 19 % of content similar to other Internet sources (excluding references). Particularly, there are 9% of content from:

https://link.springer.com/article/10.1007%2Fs10339-017-0818-5

An un-cited previous work of the authors.

Authors should cite their previous work and state which are the new additions / contributions compared to their previous work.

Also, authors should reduce similar paragraphs, as lines 39 to 48, and lines 61 to 104 are almost copied. Also, lines 531 to 538. Rephrasing or changing these paragraphs is important for the paper to be accepted. 

Some minor mistakes:

figure 1 -> Figure 1

table 3 -> Table 3. Please mind the MDPI template and editorial recommendations.

an adaptive mechanism is shown to provides -> provide

operator in order to user -> use

Author Response

We would like the thank the reviewer for their comments. Below we provide a point-by-point response to the reviewers comments and suggestions.

References are rather old, mainly from, 2007 to 2013. From 2015 to 2018 (last four years) there are only 4 journal references out of a total of 39 references. Authors are encouraged to find more current works (of high-quality journals if possible) as it seems that their approach is not compared against current works.

More recent references have been added in the paper, so our approach can be compared against current works.

Regarding Cyber-Physical systems, the term is included in the title, as an important aspect considered in the paper. However, the term is never introduced in the manuscript text. Authors should base their approach in CPS or either remove this term from the title, as it is confusing.

Taking your comment into account a paragraph has now been added in the Introduction of the paper to make the connection more clear and support the existance of CPS in the title.

The paper proposes an adaptive architecture in the title. However, I cannot see in which part of the text is the architecture proposal. What is your concept of architecture? Doesn’t it involve any architecture diagram (functional architecture)?

An ESN diagram has been added. The adaptation rules of the reservoir together with the formulation of the procedure, the user responding to the robot’s movements, complete the proposed architecture.

The experimental setup and result evaluation are correct and well-performed.

A similarity check by using the Turnitin tool results in 19 % of content similar to other Internet sources (excluding references). Particularly, there are 9% of content from:

https://link.springer.com/article/10.1007%2Fs10339-017-0818-5

An un-cited previous work of the authors.

Authors should cite their previous work and state which are the new additions / contributions compared to their previous work.

The citation has been added.

Also, authors should reduce similar paragraphs, as lines 39 to 48, and lines 61 to 104 are almost copied. Also, lines 531 to 538. Rephrasing or changing these paragraphs is important for the paper to be accepted.

The aforementioned parts of the paper have been changed/ rephrased, having now references to the paper mentioned above.

Some minor mistakes:

figure 1 -> Figure 1,

table 3 -> Table 3. Please mind the MDPI template and editorial recommendations.,

an adaptive mechanism is shown to provides -> provide,

operator in order to user -> use,

The minor mistakes have been corrected.

Round 2

Reviewer 1 Report

The authors have addressed all my comments.  On a side note, authors are encouraged to highlight all changes introduced in their manuscript to ease the review process in future submissions.

Author Response

We would like to thank the reviewer for their time, effort and valuable points.

Reviewer 2 Report

Authors have addressed almost all reviewers’ comments.

However, I still see a lack of relation between the title and the scope of the paper. Authors claim in the title that their contribution put us closer to the Human Driven CPS.

Introducing a paragraph to describe CPS in order to support the existence of CPS in the title is not enough. If the purpose of the paper is to achieve Human Driven Cyber-Physical systems, previous CPS initiatives must be studied, CPS requirements regarding human intervention must be analyzed, human-related requirements in CPS must be extracted, and human-in-the-loop concept must be validated in the proposed architecture.

I encourage the authors to make significant changes in key sections to introduce the term Human Driven Cyber-Physical systems as a core aspect of their paper.

Some other minor changes:

Line 41 remove “CITATION TO THE PAPER”

Author Response

We understand the reviewer's concern and we would like to propose an alternative. We also feel that making extensive changes throughout the manuscript will change its character. Thus, our proposal is to

remove the paragraph on CPS - added on the first round of review- from the introduction.

Changing the title to “Effective Behavioural Dynamic Coupling through Echo State Networks”

We hope that such a change satisfies the reviewer. The changes are present in the resubmitted manuscript.

Round 3

Reviewer 2 Report

I consider that the title change of the authors now correctly reflects the scope of the paper.

I’m satisfied with author’s response and all my doubts and remarks have been already solved.

In my opinion, the paper is ready to be accepted.